# Overcoming Alignment Constraints: G-Patch for Practical Adversarial Attacks on ViTs

## Abstract

This paper addresses the vulnerability of adversarial patches designed for vision transformers, which traditionally depended on precise alignment with patch locations. Such alignment constraints hindered practical deployment in the physical world. We propose the G-Patch, a novel method for generating adversarial patches that overcomes this constraint, enabling targeted attacks from any position within the field of view. Instead of directly optimizing the patch using gradients, we employ a sub-network structure for patch generation. Our experiments demonstrate the G-Patch's effectiveness in achieving universal attacks on vision transformers with a small size. Further analysis shows its resilience to challenges like brightness restriction, color transfer, and random noise, enhancing robustness and inconspicuousness in real-world deployments. Black box and real-world attack experiments validate its effectiveness even under challenging conditions. [1]

## 1 Introduction

Recently, vision transformers (ViTs) have garnered significant attention due to their impressive performance and their ability to surpass convolutional neural networks (CNNs) in various domains (Dosovitskiy et al., 2020; Chen et al., 2021a;b; Graham et al., 2021; Han et al., 2021; Liu et al., 2021; Touvron et al., 2021; Xiao et al., 2021). This remarkable performance has spurred interest in examining the robustness of ViTs, particularly considering the well-known vulnerability of CNNs to adversarial attacks (Bhojanapalli et al., 2021; Qin et al., 2022; Salman et al., 2022; Shi et al., 2022).

Drawing from the lessons learned with CNNs, adversarial attacks can be classified as universal or non-universal. A non-universal attack typically makes minute modifications to the source image (Goodfellow et al., 2014; Madry et al., 2017; Wu et al., 2020a). These approaches exhibit a significant drawback as they are customized for specific source images and have limited applicability in the physical domains. Moreover, vision transformers have demonstrated remarkable robustness against these types of attacks (Bhojanapalli et al., 2021), showing their resilience when facing such adversarial attacks.

In contrast, the universal attack aims to create a patch that can be put alongside the target, without prior knowledge of elements present in the scene (Brown et al., 2017). Adversarial patches have proven highly effective against CNNs in the physical world, as they can be positioned anywhere within the classifier's field of view to launch an attack.

Unlike CNNs, vision transformers treat the input image as a sequence of image patches. To carry out an adversarial patch attack, a commonly employed approach is to substitute certain input image patches with adversarial samples (Fu et al., 2022; Gu et al., 2022). These studies have shown the heightened vulnerability of vision transformers to adversarial patches. However, all the experiments conducted so far have been limited to the digital domain to accurately locate the adversarial patches, and even a slight shift of a single pixel could dramatically decrease the attack success rate (resulting in a loss of 70% attack success rate).

To overcome these strict limitations and enable physical-world deployment, we propose a novel approach that uses a sub-network structure to generate universal and targeted adversarial patches

---

[1]Source codes are provided in the supply materials.

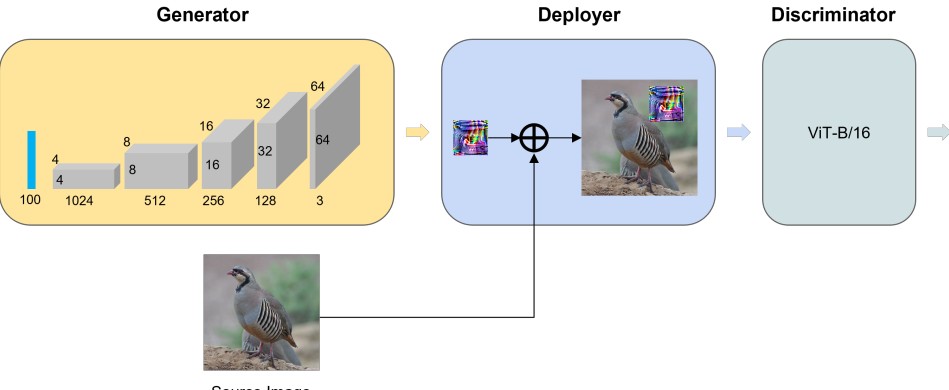

Figure 1: Network components: Generator (G-Patch generation), Deployer (random positioning), Discriminator (victim network)

(G-Patch). Our model consists of three main components: the generator, deployer, and discriminator. The generator is responsible for creating an adversarial patch. The deployer attaches the generated patch to a random position on the source image. Finally, the victim network acts as the discriminator, providing predictions based on the modified image. Notably, unlike generative adversarial network (GAN) setups, the discriminator (victim network) remains **unaltered** throughout the training process.

Our experiments demonstrate that the G-Patch can successfully launch attacks on various victim models at any position within the field of view. These patches achieve a high targeted attack success rate of over 90% while maintaining a small size of ∼12% of the source image. Further analysis reveals that the G-Patch exhibits strong robustness to brightness restriction, color transfer, and random noise. This robustness to different distributions enhances their effectiveness during physical-world deployment. To evaluate practical performance, we conducted tests on the success rate of the black box attack and the real-world placement of the G-Patch. The findings indicate that the G-Patch exhibits black box transferability and consistently demonstrates robust performance in physical environments.

Our contributions can be summarized as follows:

- We propose a new model to generate the adversarial patch for vision transformers (G-Patch), which can launch targeted attacks while overcoming the alignment constraints.

- We demonstrate that G-Patch exhibits strong robustness to brightness restriction, color transfer, and random noise, enhancing its effectiveness and inconspicuousness in physical-world deployments.

- The G-Patch is the first adversarial patch for vision transformers that can be deployed in black box and physical-world attacks.

## 2 RELATED WORK

### 2.1 VISION TRANSFORMER

The transformer was first introduced by Vaswani et al. (2017) for natural language processing (NLP) tasks. Following the success in NLP, Dosovitskiy et al. (2020) proposed the vision transformer (ViT) that leveraged non-overlapping patches as tokens input to a similar attention based architecture. Since then, numerous models have been proposed to alleviate training challenges or enhance the performance of vision transformer models. Touvron et al. (2021) introduced a teacher-student strategy in their DeiT models that dramatically reduced the pre-training request. Liu et al. (2021) proposed the SWIN transformer using the shifted windowing scheme that achieves greater efficiency by limiting self-attention computation to non-overlapping local windows while also allowing

for cross-window connection. As the vision transformer continues to advance, achieving state-of-the-art performance and becoming increasingly accessible for pre-training (Zhang et al., 2021; Tu et al., 2022; Dong et al., 2022; Zhai et al., 2022; Yao et al., 2023), it has seen widespread adoption in diverse visual tasks, including video processing (Arnab et al., 2021; Liu et al., 2022), dense prediction (Ranftl et al., 2021) zero-shot classification (Radford et al., 2021), captioning (Li et al., 2022), and image generation (Rombach et al., 2022).

## 2.2 ADVERSARIAL ATTACK

Adversarial attacks are widely employed to deceive deep learning models, resulting in remarkable successes. The first adversarial attack for computer vision tasks was introduced by Szegedy et al. (2013). Since their seminal work, numerous researchers have devised increasingly efficient techniques for generating adversarial attacks (Moosavi-Dezfooli et al., 2017; Athalye et al., 2018; Huang et al., 2019; Karmon et al., 2018; Brown et al., 2017).

In computer vision tasks, adversarial attacks can be classified into two types, depending on their reliance on the input image. The first type is non-universal adversarial attacks, which typically make minute modifications to the source image. These attacks employ various optimization strategies such as the Fast Gradient Sign Method (FGSM) (Goodfellow et al., 2014), Projected Gradient Descent (PGD) (Madry et al., 2017), and Skip Gradient (SGD) (Wu et al., 2020a). Such non-universal attacks frequently go unnoticed during deployment, diminishing their detectability by human observers across various tasks. However, these approaches often exhibit weaknesses due to their design being tailored to specific source images or limited to the digital domain.

In contrast, the second type of attack, universal attacks, uses an additional object (patch) to eliminate the requirement of relying on the input image. This patch replaces a portion of the source image and launches an attack without prior knowledge of the other items within the scene. The first universal attack approach was proposed by Brown et al. (2017). They used gradient-based optimization to iteratively update the pixel values of the patch to find the optimal values that can cause the victim model to misclassify the object (AdvPatch). The AdvPatch can be placed anywhere within the classifier's field of view, enabling attackers to craft physical-world attacks easily. Since then, many studies have followed the same strategy to develop patches for physical-world attacks aimed at deceiving classifiers or object detectors (Evtimov et al., 2017; Zhang et al., 2019; Thys et al., 2019; Wu et al., 2020b).

## 2.3 ADVERSARIAL ATTACK FOR VISION TRANSFORMER

Shortly after the introduction of the vision transformer, several researchers (Bhojanapalli et al., 2021; Shao et al., 2021; Naseer et al., 2021; Paul & Chen, 2022) conducted studies demonstrating the superior robustness of vision transformers compared to CNNs when the entire image is perturbed with adversarial perturbations (non-universal attacks). However, subsequent research by Fu et al. (2022) explored the vulnerability of vision transformers to adversarial patch attacks and found that vision transformers are more susceptible to such attacks compared to CNNs. Additionally, Gu et al. (2022) further showed that whereas vision transformers are generally resilient to patch-based natural attacks, they are more vulnerable to adversarial patch attacks when compared to comparable CNNs.

All the preceding studies used a generation method similar to the one employed for CNNs, which involves replacing certain parts of the input images with random noise and uses gradient-based optimization to iteratively update the pixel values, aiming to find the optimal values that can deceive the target model. However, unlike adversarial patches for CNNs, the patches they obtained for vision transformers must be precisely aligned with the patches used for linear projection in transformers. Gu et al. (2022) demonstrated that even a slight shift of a single pixel could dramatically decrease the attack success rate. The strict requirement posed a significant challenge to the practicality of using adversarial patches in real-world scenarios, as misalignment between attack patches and image patches is commonly encountered due to various factors. Consequently, it is crucial to develop methods for creating adversarial patches that account for these realistic conditions where misalignment can occur.

## 3 NETWORK

Instead of relying on direct gradient optimization of a random initial patch, our approach employs a sub-network to generate the desired adversarial patch from a random initial input. The resulting adversarial patch, known as the G-Patch, is affixed to a random position on the source image and then sent to the discriminator for classification. The loss, calculated based on the prediction and target class, is used to optimize the generator. The model can be divided into three functional components: the generator, deployer, and discriminator, as shown in Figure 1.

**Generator:** The generator is a sub-network consisting of five convolutional layers, each accompanied by batch normalization and ReLU activation layers. The first convolutional layer is responsible for projecting and reshaping the 100-dimensional random input vector into a three-dimensional tensor (4x4x1024 in the figure). The next four convolutional layers progressively upsample and refine the feature maps, capturing more complex patterns and details. The last convolutional layer is followed by a threshold layer instead of the batch normalization and ReLU layers. This threshold layer ensures that the output values of the generator are limited to a specific range. The threshold layer is defined as follows:

$$Th(x) = k * tanh(x) + k \tag{1}$$

where $k$ is a hyperparameter to adjust the range of the output and $tanh(x)$ applies the hyperbolic tangent function element-wise. We add $k$ here to ensure that all values in the output remain above 0.

In default, we use $k = 0.5$ to scale the range of the patch to 1, and for the following experiments, we use different $k$ to achieve brightness restriction. By changing the kernel size and stride of different convolutional layers, we can change the size of the output adversarial patch.

**Deployer:** Given an image $x \in [0, 1]^{w \times h \times c}$ with class $y$ and the generated adversarial patch $p$. We use Algorithm 1 to generate a binary mask $M$ with the same shape of $x$ at a random position.

---

**Algorithm 1** Mask generation

    **Input** source image: $x$, adversarial patch: $p$

1:  $M \leftarrow \text{Zeros}(x)$                                              $\triangleright$ all-zero mask $M$ with shape $x$
2:  $k \leftarrow \text{Randint}(0, M.shape[0] - p.shape[0])$
3:  $l \leftarrow \text{Randint}(0, M.shape[1] - p.shape[1])$             $\triangleright$ random position $(k, l)$ within $M$
4:  **for** $i \leftarrow k$ **to** $k + p.shape[0] - 1$ **do**
5:      **for** $j \leftarrow l$ **to** $l + p.shape[1] - 1$ **do**
6:         $M[i, j] \leftarrow 1$       $\triangleright$ mask only includes elements within shape $p$ at position $(k, l)$
7:      **end for**
8:  **end for**
9:  **return** $M$

---

Then the modified image is generated by the deploy function $T(p, x)$:

$$T(p, x) = M * p + (1 - M) * x \tag{2}$$

The deployer generates modified images by using the G-Patch to replace a random part of the source image, distinguishing it from prior methods in which adversarial patches replaced input patches for vision transformers.

**Discriminator:** The discriminator in our network is composed of the victim network (ViT-B/16 in the figure). It can be replaced with various models to create G-Patches tailored for different target networks or employ a combination of models to generate G-Patches for black-box attacks. Unlike GANs, the discriminator in our network remains **unaltered** throughout the training process.

For a targeted attack, the final loss of our network can be formed as follows:

$$L = log(softmax(Pr(\hat{y}|T(p, x)))) \tag{3}$$

where the $\hat{y}$ is the target class and $\hat{y} \neq y$, $Pr$ is the prediction of the discriminator with respect to class $\hat{y}$.

## 4 EXPERIMENTAL RESULTS AND ANALYSIS

In this section, we first provide detailed information about the experimental setup used in our study. Next, we show the G-Patches generated by our proposed model and evaluate their performance on various victim networks. Our results highlight the effectiveness of these patches in launching attacks from any position within the field of view. Furthermore, we conduct an analysis of the robustness of the G-Patches. We investigate their resilience to brightness restrictions, color transfer, and random noise, aiming to provide insights into their stability and effectiveness in the presence of physical environmental disturbances. Lastly, we validate the practical applicability of the G-Patch through black box attacks and by placing it in physical-world scenarios. These empirical evaluations demonstrate that the G-Patch can effectively deceive vision transformers in complex physical environments.

### 4.1 EXPERIMENTAL SETUP

In our experiments, we use the weights and shared models from the *Pytorch Image models* repository (Wightman, 2019), which have been trained on the ImageNet1K dataset. Similar to the pre-existing works, the effectiveness of the G-Patch is evaluated using white box attack settings, where the G-Patch is trained and tested on the same models. Specifically, we choose the ViT and SWIN transformer as the primary victim network architectures. These two networks exemplify crucial differences in patch handling within vision transformers: while the ViT employs fixed, non-overlapping patches, the SWIN transformer integrates varying patch sizes with shifts. We optimize our network's classification loss $L$ using the Adam optimizer ($lr = 0.001$, $\beta_1 = 0.9$, $\beta_2 = 0.999$) and conduct training for each configuration over 80 epochs, selecting the patch that achieves the highest performance as the final output patch. Additionally, input images are standardized to dimensions of 224x224, with pixel values normalized to the range of [0,1] for consistency.

To assess the attack success rate (ASR), we begin by assembling a collection of images that are accurately classified by victim models. The total number of these collected images is denoted as $P$. we apply the G-Patch to this set of images and determine the number of images, denoted as $Q$, that are classified as the target class. The ASR is then defined as $\frac{Q}{P}$, serving as a metric to measure the effectiveness of the attack.

In order to evaluate the patch's performance in the physical world, we use an HP laser printer to print the adversarial patches on A4 paper. Then we position the printed adversarial patch alongside the target object and capture photographs using a Google Pixel 6a smartphone. This physical-world test incorporates various real-world factors such as camera angle changes, lighting variations, and different types of noise.

### 4.2 PERFORMANCE OF THE G-PATCH

Table 1: Attack success rates of G-Patches on various vision transformers

| Models | Patch size | | | |
| --- | --- | --- | --- | --- |
| | 48x48 | 64x64 | 80x80 | 96x96 |
| ViT-B/16 | 6.7% | 76.4% | 97.1% | 98.7% |
| ViT-L/16 | 2.7% | 64.3% | 88.7% | 97.6% |
| SWIN-S/16 | 77.7% | 95.7% | 99.5% | 99.9% |
| SWIN-B/16 | 59.5% | 94.8% | 99.3% | 99.8% |
| DeiT-S/16 | 42.6% | 97.3% | 99.8% | 99.9% |
| DeiT-B/16 | 19.6% | 98.6% | 99.9% | 99.9% |

The summarized attack success rates (ASR) for various vision transformers, with random patch positions, are presented in Table 1. It is evident that regardless of the architecture or depth of the vision transformers, the G-Patch can achieve a high attack success rate even with a relatively small size (80x80, $\sim$12% of the input image area). Conversely, the **control patch** (natural image) with the same size achieves less than $\sim$**3%** ASR across all models. These results demonstrate the effectiveness of the G-Patch for launching attacks from any position within the field of view.

Note that the patch size needed to consistently deceive the model in this universal setting (targeted attack with random positions) is significantly larger than that required for attacks with non-targeted and aligned patches. For instance, Gu et al. (2022) show that 2% of the input image area (32x32) is sufficient to deceive a DeiT-S into misclassification. However, as observed in attacks on CNN models, non-targeted attacks are significantly easier to execute than targeted ones and require smaller patch sizes to deceive the target network. Moreover, it's essential to emphasize that the practical feasibility of these two methods varies significantly, as patches demanding precise alignment are rarely viable for real-world applications.

Some modified images created for the ViT-B/16 and SWIN-B/16 are shown in Figure 2. These images demonstrate the flexibility of our patch placement methodology, as the G-Patch is positioned randomly on the source image, regardless of its specific position or alignment.

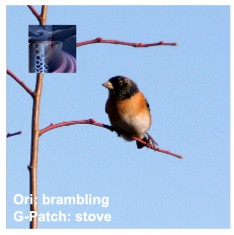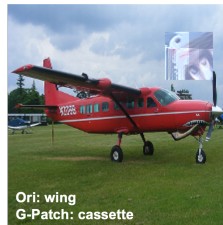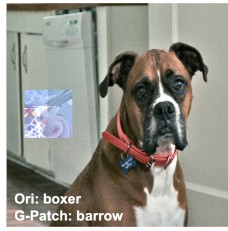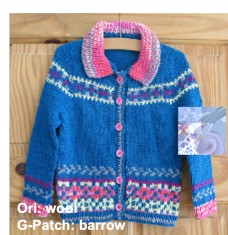
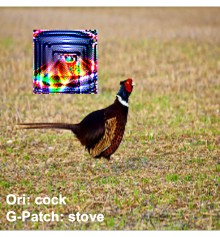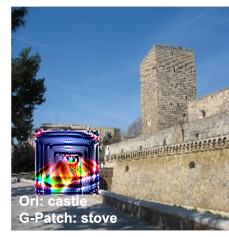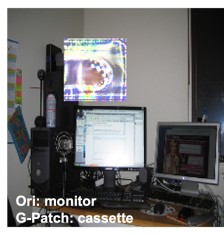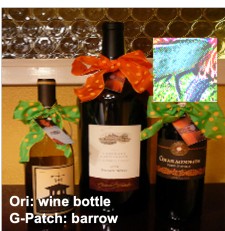

Figure 2: Modified images and predictions with different models
(Top: G-Patch on SWIN-B/16; Bottom: G-Patch on ViT-B/16)

### 4.3 PATCH ROBUSTNESS ANALYSIS

In order to evaluate the robustness of our G-Patch in real-world scenarios, we conducted an investigation into its performance when faced with challenges such as brightness restrictions, color transfers, and random noise. These aspects of robustness hold utmost significance in real-world deployments as they can substantially influence both the success rate of attacks and the visibility of patches. In this session, we assess the impact of different features on the victim networks (ViT-B/16 and SWIN-B/16) using a fixed 80x80 patch size.

#### 4.3.1 BRIGHTNESS RESTRICTION

Brightness restriction for the G-Patch can be easily implemented by adjusting the $k$ value within the threshold layer. The performance variations across different brightness ranges, as well as the brightness distributions of certain brightness-restricted patches, are illustrated in Figure 3.

The results demonstrate that G-Patch exhibits remarkable robustness to brightness restrictions. Even when the brightness range diminishes to just half of its original magnitude, the generated patch can still preserve over 80% of its ASR on a more challenging ViT-B/16 model.

The observed higher level of robustness in the SWIN-B/16 model can be attributed to the model's inherent vulnerability. It requires comparatively less information to deceive the network, as demonstrated in Table 1, where a G-Patch achieves superior performance on SWIN-B/16 with a smaller patch size.

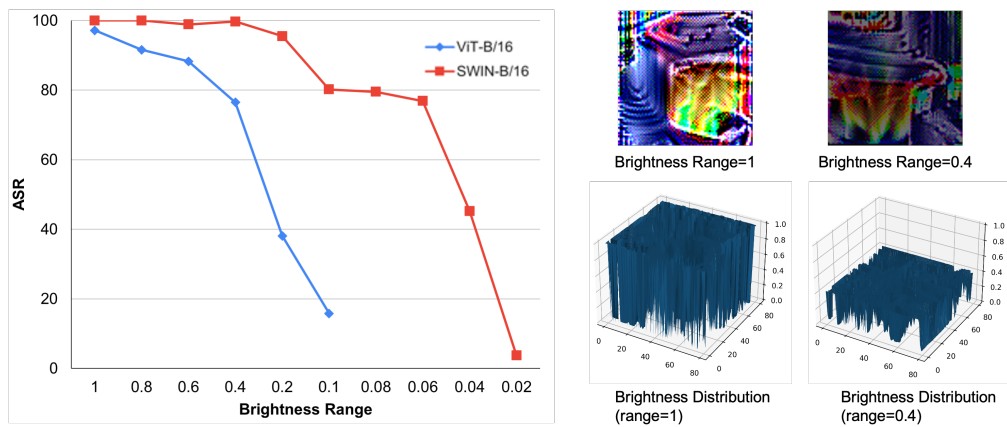

Figure 3: Attack success rates with different brightness ranges

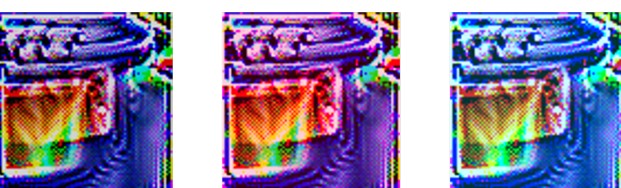

Figure 4: Different color transferred G-Patches on ViT-B/16

### 4.3.2 COLOR TRANSFER

Color transfer is a common occurrence in real-world deployments, often resulting from varying lighting conditions or shifts in printer color accuracy. To perform color transfer, we introduce a parameter $\delta$ and apply it to all values within specific channels (RGB color model). To prevent potential overflow problems while manipulating colors, we use a brightness-restricted G-Patch with a range of 0 to 0.8, as opposed to the original G-Patch, which ranges from 0 to 1. Notably, the texture distribution of the G-Patch remains unchanged throughout this color transfer process.

Table 2: Attack success rates with different color transfer

| Models | Color | | |
| --- | --- | --- | --- |
| | original color | color 1 | color 2 |
| ViT-B/16 | 91.6% | 90.9% | 90.7% |
| SWIN-B/16 | 99.6% | 98.9% | 99.7% |

We show some color-transferred patches in Figure 4, and the performance of different patches is presented in Table 2. We observe that the G-Patch consistently maintains nearly identical ASR when subjected to different color transfers, irrespective of the structure employed. This performance underscores that the G-Patch does not rely on color information to deceive victim networks.

This characteristic not only bolsters the G-Patch's robustness when deployed under varying lighting conditions but also equips it with the capability to reduce its visual conspicuousness through color transfer during deployment.

### 4.3.3 RANDOM NOISE

In real-world deployments, random noise represents one of the most prevalent challenges that an adversarial patch must confront. Such noise can originate from multiple sources, including the camera system (*e.g.*, dust on the lens, image signal processing in the camera), environmental factors (*e.g.*, fog, shadows), and even the patch itself (*e.g.*printer color accuracy, carrier texture). These challenges

create a substantial divide between digital attack results and physical-world results, rendering many digital-domain attack techniques ineffective in the physical world.

To simulate the noise commonly encountered during real-world deployment, we generate random noise based on different signal-to-noise ratios (SNR). In order to avoid overflow when adding strong noise, we choose patches with a narrow brightness range (ranging from 0 to 0.6). The results of our experiments are shown in Table 3.

Table 3: Attack success rates across different noise levels

| Models | | SNR | | | |
|---|---|---|---|---|---|
| | Ori. | 10 dB | 7 dB | 5 dB | 4 dB |
| ViT-B/16 | 86.5% | 85.6% | 83.2% | 69.3% | 58.6% |
| SWIN-B/16 | 99.3% | 97.4% | 83.6% | 67.6% | 44.2% |

We observe that the performance of the G-Patch remains relatively stable even at an SNR of 7 dB (20% random drift). The patch's ability to maintain a high attack success rate in the presence of such noise further reinforces its effectiveness and practicality.

## 4.4 BLACK BOX ATTACK

Conducting black box attacks in real-world deployments is an exceptionally challenging task. Prior adversarial patches designed for vision transformers have shown very limited black box attack transferability (ASR less than 5%). To evaluate the black box attack performance of our G-Patch, we train the patch on ViT-B, SWIN-S, and SWIN-T, and evaluate it on SWIN-B. The corresponding attack success rates are summarized in Table 4.

Our observations indicate that the G-Patch requires a larger size to deceive a vision transformer under a black box attack setting. However, it remains considerably more efficient than the control patch (a natural image) of the same size. Furthermore, when comparing its performance with that of the AdvPatch employed in CNNs, as demonstrated in (Brown et al., 2017), the G-Patch achieves a comparable attack success rate on vision transformers of equivalent size.

Table 4: Black box attack success rates

| Patch | Patch size | | |
|---|---|---|---|
| | 80x80 | 96x96 | 112x112 |
| G-Patch | 50.4% | 76.4% | 83.2% |
| Control patch | 2.7% | 3.9% | 6.8% |
| AdvPatch (CNNs) | 58% | 81% | 86% |

## 4.5 PHYSICAL-WORLD ATTACK

The physical-world deployability is a key feature that makes adversarial patches more popular than many other attacking methods. However, due to the alignment problem, none of the adversarial patches designed for vision transformers has been deployed in the physical world before. Although our G-Patches show the perfect position irrelative based on previous experiments, a valid concern remains regarding their robustness in physical-world scenarios. In order to address this concern, we design several physical-world deploy instances to show that the proposed attack patch can still work robustly in the physical world.

We have chosen a variety of scenarios encompassing different lighting conditions and image captures from varying distances and angles. For our experiments, we use the ViT-B/16 model as the victim network, and we present some illustrative figures and predictions in Figure 5.

The top row shows the G-Patch's remarkable effectiveness in addressing distortions resulting from variations in camera angles, shadows, and even the bending of the printed patch. The bottom row highlights the G-Patch's ability to deceive the network with a relatively small size compared to the

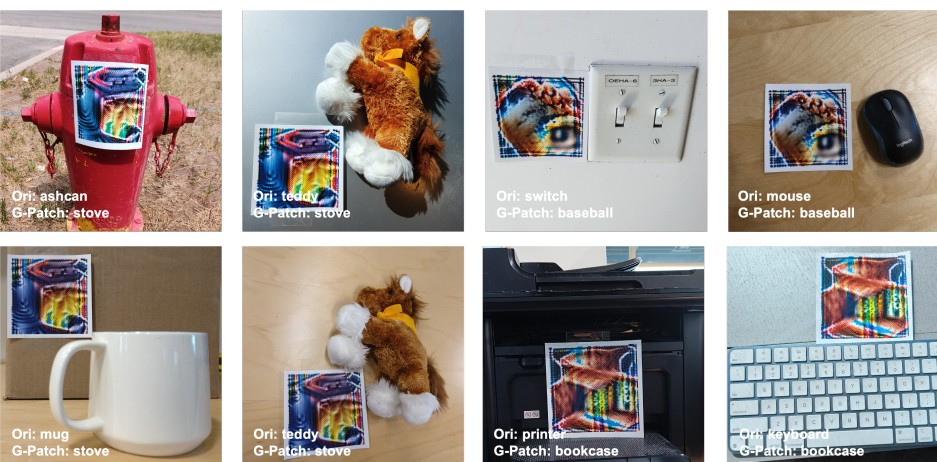

Figure 5: Prediction results in the physical world

original target. These results demonstrate the robustness of G-Patch in launching attacks in the complex physical world, addressing practical challenges encountered when designing adversarial patches for vision transformers.

## 5 CONCLUSION

This paper introduces the G-Patch, a novel universal adversarial patch designed for vision transformers that can deceive them without alignment constraints. Our experiments demonstrate the effectiveness of the G-Patch, allowing it to launch attacks from various positions within the field of view with a relatively small size. Additionally, we highlight its robustness against brightness restrictions, color transfer, and random noise, making it resilient in real-world scenarios. Black box and physical-world attack experiments validate its effectiveness under challenging conditions. The G-Patch represents a significant advancement, bridging the practicality gap between the digital and physical domains for adversarial patches on vision transformers, opening avenues for further research.

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
