# OpenReview forum: "Overcoming Alignment Constraints: G-Patch for Practical Adversarial Attacks on ViTs"
_ICLR.cc/2024/Conference — ICLR 2024 Conference Withdrawn Submission_

### Official Review · Reviewer_EEsX · 2023-10-27

**Soundness:** 3 good
**Presentation:** 3 good
**Contribution:** 3 good
**Rating:** 5
**Confidence:** 4

**Summary:**

This paper proposes a novel method called G-Patch for generating adversarial patches for vision transformers (ViTs) that overcomes the alignment constraints of previous methods. The authors demonstrate the effectiveness and robustness of G-Patch in achieving universal targeted attacks on ViTs, even in challenging conditions. The experiments show that G-Patch can be deployed in black box attacks and in real-world scenarios. The paper makes several contributions by introducing a new model for generating adversarial patches for ViTs, demonstrating the robustness of G-Patch to various challenges, and showing its effectiveness in black box and physical-world attacks.

**Strengths:**

(1) The proposed G-Patch method is innovative and addresses the alignment constraint issue associated with previous adversarial patch methods for ViTs.

(2) The experiments conducted demonstrate the effectiveness of G-Patch in achieving high targeted attack success rates while maintaining a small patch size.

(3) The analysis of G-Patch's robustness to brightness restriction, color transfer, and random noise enhances its practicality and effectiveness in real-world deployments.

**Weaknesses:**

(1) The paper lacks in-depth theoretical analysis of the proposed method. More detailed explanations of the sub-network structure used for patch generation would be beneficial.

(2) The paper lacks a comprehensive comparison with widely-known baselines in the field of adversarial attacks on ViTs. It would be helpful to include comparisons with existing patch generation methods for ViTs.

(3) For real-world synthesizing, I think the motion-level corruptions would be more helpful, see [1]

(4) For outdoor scenes, I think the patches are too big.

[1] Benchmarking Robustness of 3D Object Detection to Common Corruptions, CVPR 2023

**Questions:**

/NA

**Details Of Ethics Concerns:**

/NA

---

### Official Review · Reviewer_sysB · 2023-10-27

**Soundness:** 2 fair
**Presentation:** 2 fair
**Contribution:** 2 fair
**Rating:** 5
**Confidence:** 4

**Summary:**

This paper explores a white-box targeted adversarial patch attack against the VIT model, G-Patch. G-Patch is robust to position, brightness, and noise. The authors conduct experiments in both digital and physical domains to verify the effectiveness of the method.

**Strengths:**

1. Targeted adversarial patch attacks can be realized with a relatively small adversarial patch area.
2. There are physical world experiments.

**Weaknesses:**

1. The description of the method section is too simple and does not explain why the method is better adapted to the VIT model. In other words, the same method is still adapted to CNN.
2. White-box performance comparisons do not compare other methods, making it difficult to demonstrate the superiority of one's own method.
3. Lack of methods under the same settings to compare black-box performance. For example, AdvPatch in Table 4 can be trained based on VIT to compare performance with G-patch.

**Questions:**

In the authors' experiments, SWIN is less robust than VIT for the same patch size, is there any reasonable explanation?

---

### Official Review · Reviewer_Bcct · 2023-10-31

**Soundness:** 2 fair
**Presentation:** 2 fair
**Contribution:** 1 poor
**Rating:** 3
**Confidence:** 4

**Summary:**

This paper proposes the G-Patch, a method for generating adversarial patches for vision transformers that overcomes the alignment constraints traditionally associated with such patches. The G-Patch is generated using a sub-network structure and can be targeted from any position. The paper demonstrates the effectiveness of the G-Patch and shows its resilience to challenges such as brightness restriction, color transfer, and random noise. Black box and real-world attack experiments validate its effectiveness under challenging conditions.

**Strengths:**

The paper presents a novel method for generating adversarial patches for vision transformers that overcomes alignment constraints.

The experiments include black box and real-world attack scenarios, demonstrating the practical applicability of the G-Patch.

**Weaknesses:**

Lack of innovation: I believe this paper lacks necessary novelty, both in motivation and in technical aspects. The pipeline designed in this paper for generating adversarial patches has been used in many similar works, such as [1][2]... Moreover, the adversarial patch designed for ViT lacks a reasonable technical design. Why is it effective for Transformer-based architecture? I did not see any targeted design in the methods section.

Overclaims: The author promotes that G-Patch has good inconspicuousness. However, from the actual visual results (in the cases of physical experiments), I don't find it to be very imperceptible. Furthermore, the author claims that the method has strong robustness against brightness constraints, color transfer, and random noise, yet these experiments lack comparative baselines.

Lack of ablation studies and comparative baselines: Although this method is an adversarial patch for the ViT architecture, the experiments lack comparisons with state-of-the-art adversarial patch methods. Additionally, there is a lack of ablation analysis, such as the impact of changes in the generator architecture.

[1] Perceptual-sensitive gan for generating adversarial patches In AAAI
[2] Generative dynamic patch attack. In BMVC

**Questions:**

see the weakness

---

### Official Review · Reviewer_u7HF · 2023-11-01

**Soundness:** 2 fair
**Presentation:** 2 fair
**Contribution:** 1 poor
**Rating:** 3
**Confidence:** 5

**Summary:**

This paper explores the adversarial robustness of Vision Transformer on adversarial patches. The authors believe that existing patch attacks on ViT will have alignment constraints, thereby limiting attack performance. Therefore, the authors proposed G-Patch to improve performance. The authors proposed a generator and random strategy for patch generation. Experiments show that the proposed method is effective.

**Strengths:**

It is meaningful to explore the adversarial robustness of adversarial patches on different architectures.

**Weaknesses:**

1. The motive is unclear. What are alignment constraints? More explanation and references are needed here.

2. Moving adversarial patches leads to a decrease in attack performance. This phenomenon [1] is seen in both CNN and ViT. Why should we focus specifically on attacking ViT?

3. In real-world scenarios, we don’t know whether DNNs are CNN or ViT. So, can adversarial patches for ViT be effective on CNN? This requires more experiments to prove.

4. Novelty is limited. Generator-based architectures [2] have been around for a long time in this field, but the authors claim that this is a new model.

5. This work is not the first adversarial patch on ViT. Prior to this work, DevoPatch [3] studied the adversarial patch performance in a black-box setting.

[1] LaVAN: Localized and visible adversarial noise, ICML 2018

[2] Perceptual-Sensitive GAN for Generating Adversarial Patches, AAAI 2019

[3] Query-Efficient Decision-based Black-Box Patch Attack, IEEE Transactions on Information Forensics and Security

**Questions:**

See Weaknesses.

**Details Of Ethics Concerns:**

The authors do not describe the potential threats posed by G-Patch in the main body.